# A Novel Multi-Candidate Multi-Correlation Coefficient Algorithm for GOCI-Derived Sea-Surface Current Vector with OSU Tidal Model

He Cui [1,2], Jianyu Chen [1,2,*], Zhenyi Cao [1], Haiqing Huang [1] and Fang Gong [1]

1   State Key Laboratory of Satellite Ocean Environment Dynamics, Second Institute of Oceanography, Ministry of Natural Resources, Hangzhou 310012, China
2   Institute of Physical Oceanography and Remote Sensing, College of Oceanology, Zhejiang University, Zhoushan 316021, China
*   Correspondence: chenjianyu@sio.org.cn

**Abstract:** The maximum cross-coefficient (MCC) algorithm based on the template matching technique is a typical algorithm for obtaining the sea-surface currents (SSCs) in marginal seas. However, this algorithm has mismatches between images in highly turbid water. In this study, we implemented the MCC algorithm to Geostationary Ocean Color Imager-derived total suspended matter to obtain the SSCs in the Yellow Sea and the East China Sea. We propose a novel vector optimization algorithm, which is combined with the accurate estimate of tidal ellipses from the OSU tidal model. This method considers the three greatest candidate acquisitions from multi-correlation coefficients as potential vectors. The rotation direction of the vector within the tidal oscillation is used to identify and substitute for the spurious vector. The obtained average speed of SSC reached 0.60 m/s, which was close to the buoy-measured average speed of 0.58 m/s. Compared with the existing spurious vector eliminating method, the average angular error was improved by 20%, and the average relative amplitude error was improved by 4% in our case study. On the basis of ensuring data integrity, the inversion accuracy was improved.

**Keywords:** geostationary ocean color imager (GOCI); OSU tidal model; multi-candidate multi-correlation coefficient algorithm; sea-surface current inversion





## 1. Introduction

Sea-surface current (SSC) is one of the most important physical properties in ocean dynamics, and is critical for understanding ocean physical and biogeochemical processes [1]. High-frequency and high-spatial-resolution current observations in nearshore water provide essential data for navigation, maritime rescue operations and environmental monitoring (such as harmful algal blooms, harmful substances and sediment transport) [2].

To monitor and forecast short-time-scale SSC change in real-time, it is necessary to use high-spatial-temporal resolution satellite data to estimate the SSC field. Some studies have shown that the high-frequency dynamic characteristics of the SSC field in a region can be obtained by using the continuous observation data of a fixed sea area from a geostationary satellite and an inversion algorithm [3–6]. As a successful pioneer, the world's first stationary ocean and water color satellite communication ocean and meteorological satellite (COMS) was launched by South Korea on June 27, 2010, carrying the geostationary ocean color imager (GOCI) [7]. Its continuous satellite optical images in Northeast Asia have been successfully applied to the inversion of SSCs in its observed sea area [8]. Choi et al. [9] used the GOCI data to estimate suspended sediment movement along the west coast of Korea; Yang et al. [10] used the GOCI data to retrieve high-frequency SSCs around the Korean Peninsula. Compared to polar-orbiting satellites, GOCI's unprecedented high spatiotemporal observations greatly improve our ability to monitor changes in the ocean's highly dynamic environment [11–13].

Different types of satellite remote sensing technologies have allowed many practical methods for monitoring SSCs. Among them, two have been most widely used: one is an algorithm based on the heat flow or tracer conservation equation [14–16], and the other is an algorithm based on the tracer feature (i.e., the maximum cross-correlation algorithm, MCC) [11,17–20]. After Emrey et al. [18] proposed that the MCC algorithm can be used to estimate the SSC field, many scholars have successfully inverted the SSC field in different sea areas by using thermal infrared remote sensing images [11,19,21] or water color remote sensing images on the basis of their algorithm [10,22]. Furthermore, recent studies have shown that water color data can better track the movement of water masses [8]. Zhu et al. [23] used the Himawari-8 data to estimate the coastal currents in Hangzhou Bay using the generalized Hough transform (GHT) method and the MCC method, respectively. The results showed that the Himawari-8 data can be used to effectively estimate the currents, and the error in the current measured using the GHT method is smaller than MCC method in the Yangtze estuary and offshore areas. Sun et al. [24] used the MCC method and robust optical flow method to process GOCI images for quantifying high-resolution, near real-time SSCs and current features. Yang et al. [10] showed that the MCC method can be applied to GOCI-derived total suspended matter (TSM) and chlorophyll-a (Chl-a) to estimate the fast tidal currents along the west coast of Korea. Lang et al. [25] used Level-1B data measured by the GOCI to identify ice pixels in the northernmost part of the Bohai Sea (Liaodong Bay), and then used the MCC method to estimate ice drift. Lou et al. [4] applied the MCC method to Rayleigh-corrected reflectance from the GOCI measurements to derive SSCs along the Zhejiang coast of the East China Sea, but they did not provide validation results.

Although these studies have shown that the use of the GOCI data and the MCC algorithm can better invert the variation characteristics of the SSC field in the target area, there are still some spurious currents in these inversion results. The MCC method has been applied to improve the accuracy of GOCI-derived SSCs in recent studies. Chen et al. [26] used the MCC method to invert the SSCs in the East China Sea and then proposed a current field vector data-processing method based on angular limitation. The average angular error (AAE) value of the processed SSC field data decreased by 28–38%. Jiang et al. [27] derived the Bohai SSCs from the GOCI data using the MCC method, and evaluated the results using the Oregon State University (OSU) tidal model and high frequency (HF) radar observations, showing that for a time difference of 1 h, a template window size of 10–15 km, and a search range of 4–8 km give the optimal results. Hu et al. [28] integrated multivariate optimum interpolation (MOI) with the MCC, and the average absolute differences between the "actual" velocity of the derived model and the velocity derived from sea-surface temperature (SST) were reduced by 19% in relative magnitude and by 22% in direction. Previous studies lacked consideration of the method itself, and the MCC method is a template-based matching technique. The reason for this method to derive spurious SSCs is that the nonlinear deformation and movement of water masses and the small-scale dynamic process in the template window may cause the image mismatch in the MCC method in high-turbidity water [29]. Furthermore, the rapid settlement and resuspension of suspended solids in high turbidity water will also affect the precision of SSC field inversion results [10].

SSCs can also be calculated by using ocean models. OSU developed the China Sea Regional Tidal Model TPXO-CSI2016 (China Seas & Indonesia 2016), which covers the South China Sea, East China Sea, Yellow Sea, Bohai Sea, and Northwest Pacific, and assimilates a large amount of satellite remote sensing data and tide gauge data with a spatial resolution of $1/30°$ [30,31]. This model can be used for tidal prediction or tidal current calculation in specific regions. Hu et al. [32] calculated the M2 tidal current using the OSU regional tidal model, and the overall OSU-derived, GOCI-derived, and measured results exhibited better consistency. Zhao et al. [33] evaluated the accuracy of seven global/regional tidal models by using the harmonic constants and tidal heights of eight main tidal components of 33 tidal stations in the coastal area of Zhejiang, and considered the model to have high accuracy in the study area. Cui et al. [34] studied the applicability of GOCI inversion

SSCs and OSU model SSCs in the Yellow Sea and concluded that the OSU model had high accuracy in simulating the current direction of offshore high-turbidity water.

We combine the GOCI data with in situ data to study the accuracy of GOCI-derived SSC field in this paper. To improve the MCC algorithm, we propose a novel multi-correlation coefficient optimization algorithm for GOCI inversion of SSC field vectors based on the tidal ellipse and OSU tidal model. We determined the rotation direction using the tidal vector to identify the spurious vector and substituted it with the appropriate vector to improve the inversion precision. The remainder of this article is arranged as follows. Dataset and methodology used are described in Sections 2 and 3, respectively. Main results and discussion are given in Sections 4 and 5, respectively. Conclusions are provided in Section 6.

## 2. Data Set

### 2.1. In-Situ Data

The data collected by the drifting buoys used cover the periods from 3 June to 20 August 2012 and from 28 July to 26 August 2013. The longitude range of buoys movement was from 120°E to 127°E, and the latitude range was from 29°N to 39°N. The trajectories of the drifting buoys illustrated in Figure 1 show they are mainly concentrated in the East China Sea, the west part of the southern Yellow Sea and the west coast off the Korean Peninsula. The position data measurement interval was approximately half an hour, and in approximately 77 days of the battery life cycle, 6505 profiles were obtained.

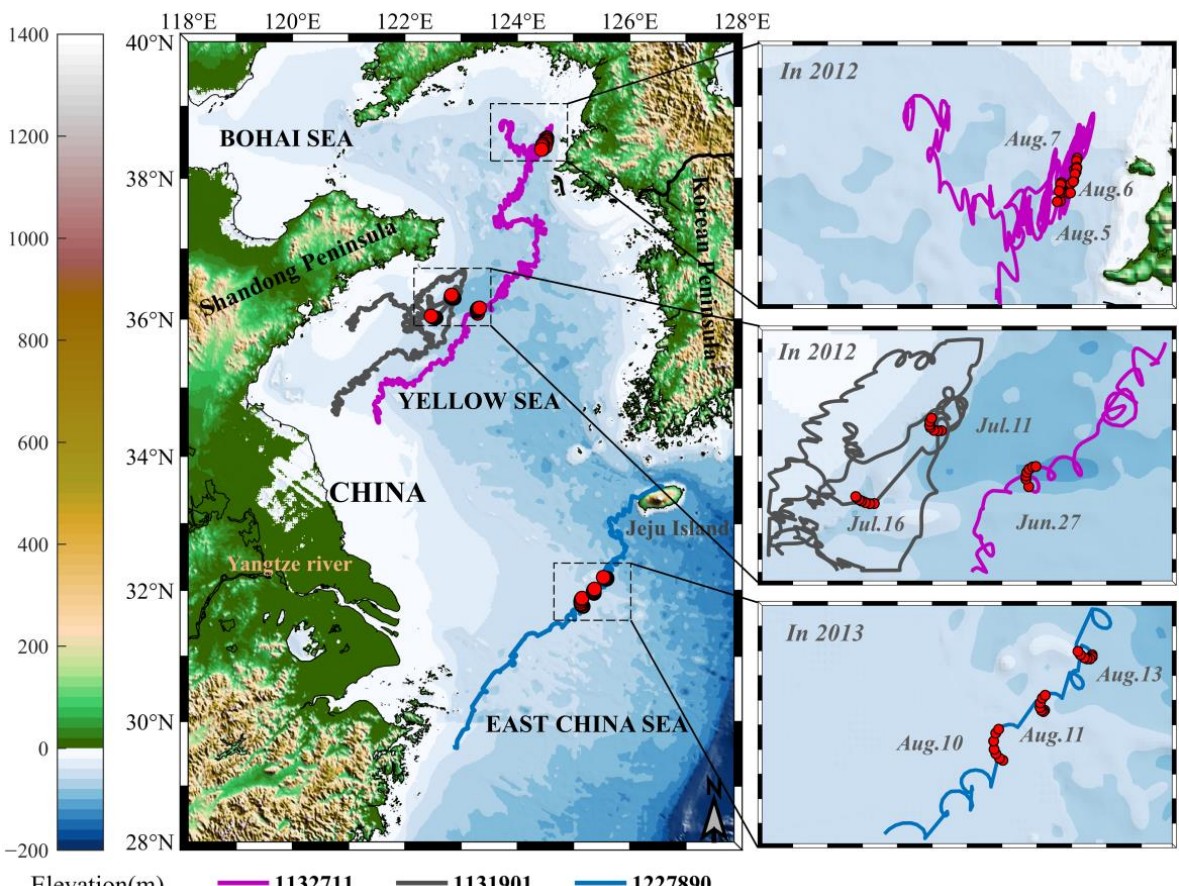

**Figure 1.** The study area and its bathymetry. From light blue to dark blue, the water depth ranges from 10 to 200 m. The color lines show the tracks of three drifting buoys. Red dots represent the buoy locations of selected cases, which are distributed in the northern Yellow Sea, southern Yellow Sea, and East China Sea from north to south.

The drifting buoy moves with the current on the sea surface or at a certain depth, which is one of the convenient and effective tools for ocean observation. It is affected by wind and currents. The principle of measuring current is to use the Lagrange method to describe the movement of sea water. The accuracy of measurement is mainly related to the ratio of the area of drifting buoy above and below the sea surface. Figure 2 is a diagram showing a working drifting buoy. The drifting buoy was similar to Davis's drogue drifter [35]. The area of the buoy on the sea surface is 0.031 m$^2$, and the area of the buoy under the sea surface is (0.031 + 0.7) m$^2$. The ratio of the upper and lower areas was as low as 0.04. The center point of the sail below the sea surface was 3 m underwater. The structure of the drifting buoy shows that the movements of the drifting buoy are mainly controlled by SSC.

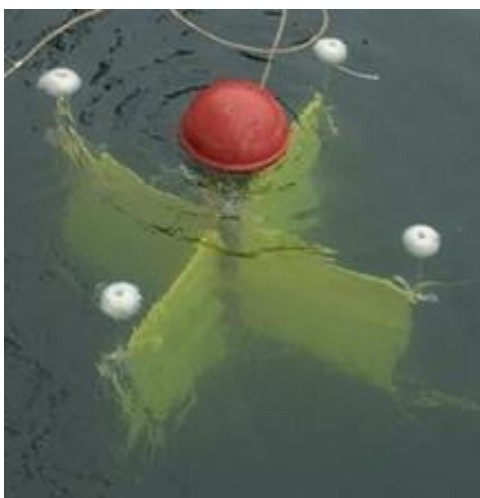

**Figure 2.** A drifting buoy at work.

### 2.2. GOCI Data

The GOCI data used in this study cover a total of eight bands of visible and near-infrared wavelengths from 402 to 885 nm. The satellite can perform hourly observation from 8:30 a.m. to 3:30 p.m. (Beijing time). Its temporal resolution (1 h) and spatial resolution (500 m) are the advantages that can be used for coastal marine environmental monitoring. GOCI level-1B (L1B) data can be downloaded from the Korea Ocean Satellite Center (http://kosc.kiost.ac.kr/, accessed on 13 July 2021) [36]. Eight groups of data can be obtained every day. There are many GOCI products, such as TSM, Chl-a, and normalized water-leaving radiance, which can be taken as tracers. Among them, in coastal areas, the current inversion using TSM as a tracer can capture the changes in tidal phases [10], and are used to compare with other bio-optical remote sensing products. The inversion results using suspended matter products are more accurate and reliable [4,29]. We use the GOCI-derived TSM as the tracer in this study.

### 2.3. OSU Tidal Current Model Data

The tide model driver (TMD) is used to run the OSU tidal current model ( http://www.tpxo.net, accessed on 13 July 2021) to obtain the u and v components of the tidal current at a specific latitude, longitude, and time, thus producing the corresponding tidal current data [37]. Using the tidal current model, we extract the tidal current data at the same observation time as the GOCI (composite tidal current data superimposed by eight main tidal components), and match them to the same 0.15° × 0.15° grid as the GOCI inversion results for comparison.

## 3. Methodology

### 3.1. GOCI Data Processing

Supported by GOCI Data Processing System software (GDPS), the GOCI-L1B original data were preprocessed in terms of atmospheric correction and masking. Then, we used the built-in TSM inversion algorithm provided by GDPS processing software to process data into an L2-level TSM product [38]. Figure 3 shows the MCC method based on the image matching method. By performing this operation, hourly SSC can be obtained from two consecutive GOCI images on the same day. The former image used to estimate the present position is called the "template window," and the latter is called the "search window." The MCC method uses correlation relationship based on template matching technique to track change in the tracer structure. Template matching is a technique for finding the most similar part of an image to another template image [39]. The template is a small image, and template matching is a search for a target in a large image, where the target is known to be in the image and has the same size, orientation and image elements as the template, and a certain algorithm can find the target in the image and determine its coordinate position. Equation (1) is used to calculate the correlation coefficient between the "template window" and the "matching window."

$$\rho\left(S_{sub}^{i}, T_{sub}^{i+1}\right) = \frac{\text{cov}(S_{sub}^{i} - T_{sub}^{i+1})}{\sqrt{\text{var}(S_{sub}^{i}) \times \text{var}(T_{sub}^{i+1})}} \tag{1}$$

where $i$ is the time scale of GOCI observations, its value of 0 to 7 representing Beijing time from 8:30 to 15:30 (1-hr interval), and $S_{sub}^{i}$ and $T_{sub}^{i+1}$ are the two-dimensional matrix data of "template window" and the search sub-window with the same size in "Search window," respectively. $\rho$ is the cross-correlation coefficient, and its variation range is [–1,1]. The closer the correlation coefficient is to 1, the more accurate the matching of the two remote sensing images is, and the more accurate the retrieved current is. Using Equations (2) and (3), the speed and direction of the current can be calculated.

$$V = \frac{\sqrt{(x_{i+1} - x_i)^2 + (y_{i+1} - y_i)^2}}{h} \tag{2}$$

$$D = \arctan2(y_{i+1} - y_i, x_{i+1} - x_i) \tag{3}$$

where $x_i$ and $y_i$ are the central coordinates of the template window, and $x_{i+1}$ and $y_{i+1}$ are the central coordinates of the matching window. $h$ is the observation time interval between $S_{sub}^{i}$ and $T_{sub}^{i+1}$, which is 1 h.

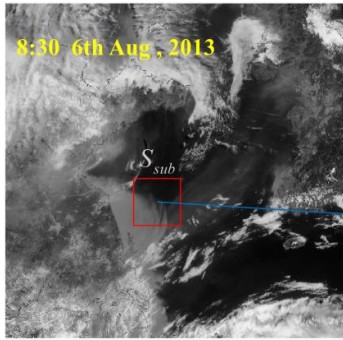 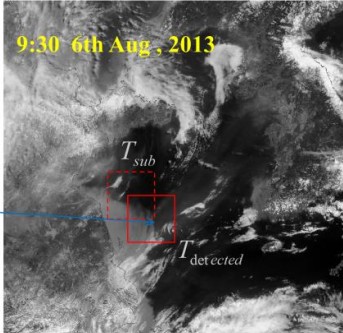

**Figure 3.** A schematic diagram of the MCC algorithm for estimating SSC field. Taking the first invertible current field as an example, the images on the left and right represent satellite remote sensing images observed in the same sea area and on the same day at 8:30 and 9:30, respectively. The solid box in the left panel is the template window, the solid box in the right panel is the matching window, and the dashed box is the same template window as that in the left panel.

The template window size selected here is $20 \times 20$ pixels, the search window size is $36 \times 36$ pixels, and the fitting threshold is set to 0.9 to obtain more accurate vectors [9,29]. The appropriate matching window is determined by calculating the correlation coefficient between the template window and the moving search window. If the calculated value is greater than the threshold value, the position that the template window moves to in unit time is considered as the matching window. Then, the above steps are repeated to obtain a relatively complete SSC field.

### 3.2. Drifting Buoy Data Processing

The information contained in the drifting buoy data is the specific time, latitude and longitude of each station. First, we calculate the distance between two continuous stations in the time series data of drifting buoys. Then the speed of the vector formed by the two points is calculated by using the time difference between the two points. For the calculation of the angle, any point is taken so that its projection with the remaining two points in the plane forms a right triangle, as shown in Figure 4. According to the distance formula, the distance between any two points can be calculated, and finally, the magnitude of the angle can be calculated according to the cosine theorem. The specific formula is as follows:

$$C = \arccos(\cos(90 - B_{\text{lat}}) \times \cos(90 - A_{\text{lat}}) + \sin(90 - B_{\text{lat}}) \times \sin(90 - A_{\text{lat}}) \times \cos(B_{\text{lon}} - A_{\text{lon}})) \tag{4}$$

$$L = R \times C \times \pi / 180^{\circ} \tag{5}$$

where $R$ is the radius of the Earth, $A_{\text{lat}}$ and $A_{\text{lon}}$ represent latitude and longitude of point A, respectively, and $B_{\text{lat}}$ and $B_{\text{lon}}$ represent the latitude and longitude of point B, respectively. C represents the radian corresponding to the angle formed by points A and B. L is the arc length corresponding to the radian C, which is the distance between points A and B.

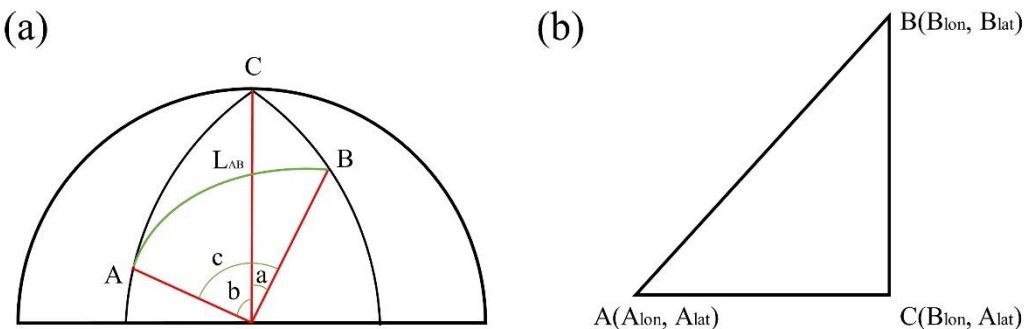

**Figure 4.** (**a**) A vector diagram formed by calculating the buoy points (A, B, and C represent the three points on the projection surface of the sphere, and the angle of the "arc" on the sphere at that point, respectively; a, b, and c represent the arc of the ABC three-point pair, O is the center of the sphere). (**b**) A right triangle composed of three points A, B, and C and the coordinates of each point.

### 3.3. Multi-Candidate Multi-Correlation Coefficient Optimization Algorithm

The prevailing tidal current is one of the important hydrodynamic processes in the Chinese seas. The precision of SSC field inversion using the remote sensing technique largely determines the accuracy of the SSC field, so it is necessary to improve the inversion precision. The OSU model current field is consistent with the in situ SSC field in terms of the current direction and can well reflect the vector rotation direction in a tidal current elliptic period. Therefore, combining the GOCI inversion SSC field and OSU model current field, we propose a multi-correlation coefficient optimization algorithm for the remote sensing inversion SSC vector based on the OSU tidal model (Figure 5). The goal is to identify and substitute the spurious vector in the GOCI inversion current field and improve the SSC precision. The main step is to obtain the three largest candidate SSC vectors under the first three correlation coefficients for seven consecutive periods of the GOCI

day by the MCC method. Figure 6a shows a schematic diagram for estimating the SSC field by the multi-correlation coefficient inversion algorithm. Based on template matching technique, this algorithm takes into account the mismatch phenomenon of this technique in high-turbidity sea areas; that is, the highest correlation coefficient does not necessarily correspond to the optimal matching and obtains the SSC vectors under the first three correlation coefficients to improve the optional rate. Then, the OSU tidal model is used to obtain corresponding tidal current vectors, and the two current field vectors are sorted according to the time order. For the sorted tidal current vectors, the rotation angle of the adjacent tidal current vectors is calculated, and the direction of the rotation angle is taken as the rotation direction. As shown in Figure 6b, the angle $a_{n-1}$ is the rotation for vectors $V_n$ and $V_{n-1}$, and the angle $a_n$ is the rotation for vectors $V_n$ and $V_{n+1}$. The direction from vector $V_{n-1}$ to $V_n$ is clockwise, and that from vector $V_{n-1}$ to $V_n$ is counterclockwise. Then, the rotation direction of the GOCI SSC vector obtained by the 1st correlation coefficient is taken as the main rotation direction, the rotation direction and rotation angle of the two adjacent vectors are calculated, and the spurious vector is determined by comparing with the tidal current vector. As shown in Figure 6c(I), $V_{n-1}$, $V_n$ and $V_{n+1}$ are the SSC vectors of three adjacent time periods in the main rotation direction, and $v_{n-1}$, $v_n$ and $v_{n+1}$ are the corresponding tidal current vectors. To facilitate the calculation of rotation angle and direction, the starting points of all vectors are placed at the same point. As illustrated in Figure 6c(II), $a$ is the rotation angle of $V_n$ and $V_{n+1}$, and $a'$ is the rotation angle of $v_n$ and $v_{n+1}$. The corresponding rotation directions of the two are opposite. It can be determined that vector $V_{n+1}$ is the spurious vector. Finally, the spurious vectors are substituted by the candidate vectors of the 2nd and 3rd correlation coefficients of the corresponding time period; and the rotation angle between the substituted vector and its adjacent vector is calculated. The vector closest to the rotation angle of the tidal current vector in the corresponding time period is selected as the optimal substituted vector to obtain the final SSC vector in the main rotation direction and improve the precision of the inversion SSC field.

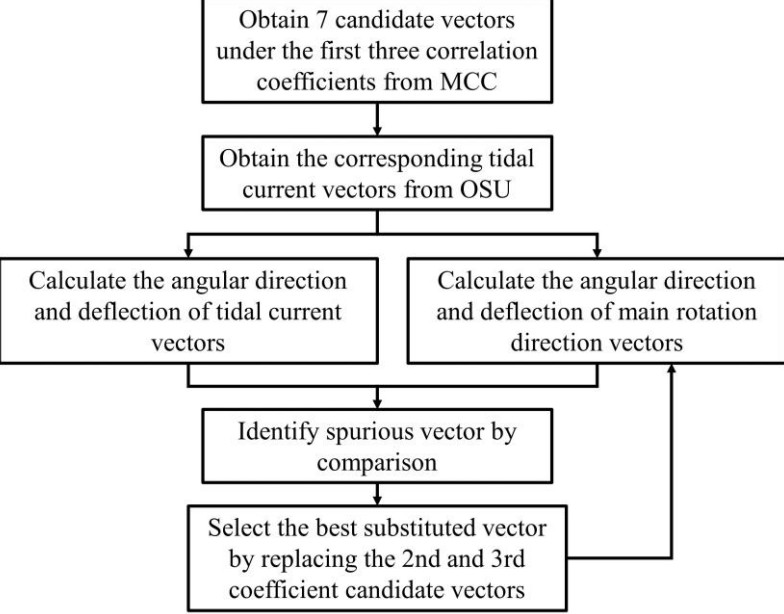

**Figure 5.** Workflow of the OSU-based multi-correlation coefficient algorithm to identify and substitute spurious vectors. MCC: Maximum cross-coefficient.

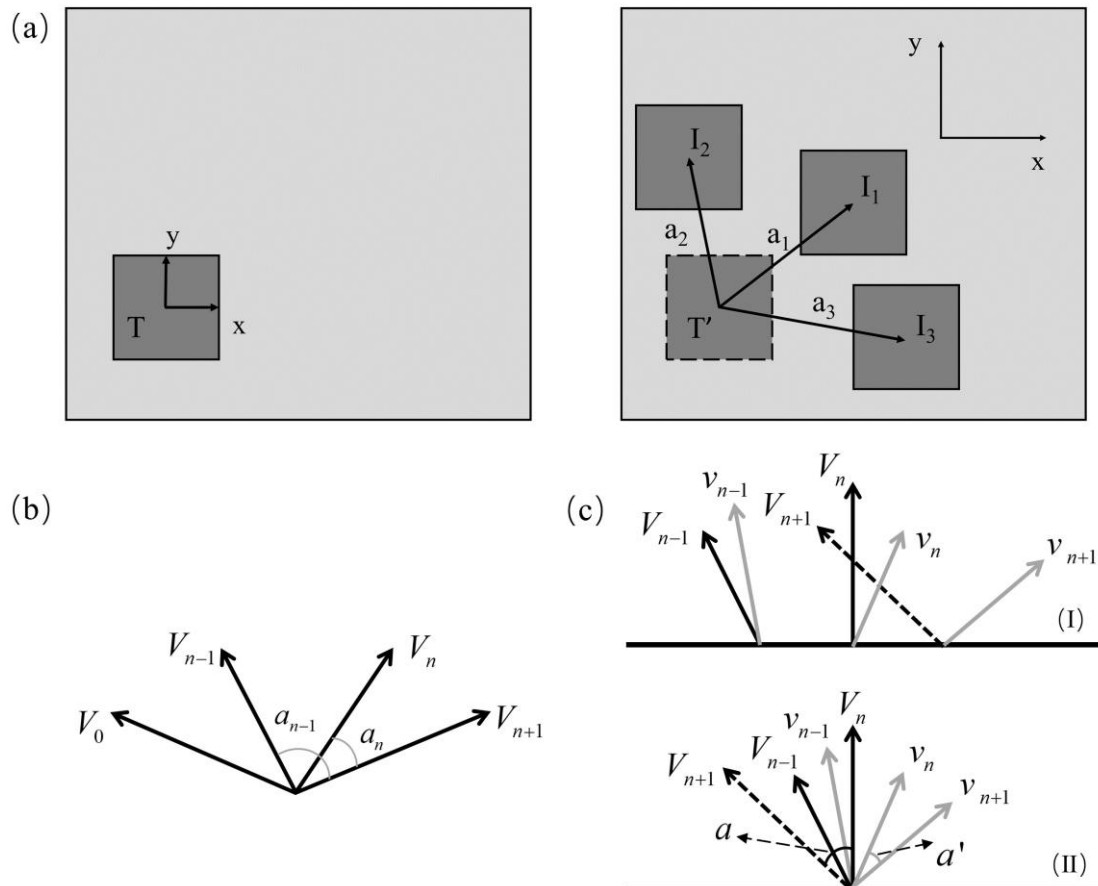

**Figure 6.** (**a**) An illustration of SSC vector estimation by multi-correlation coefficient inversion algorithm. $T$ is the matching template, $T'$ is the mapping of the matching template at the same position in the search area, $I_1$, $I_2$ and $I_3$ are the regions searched by the matching template, and $a_1$, $a_2$ and $a_3$ are the current field vectors under the corresponding correlation coefficients; (**b**) an illustration of vector direction judgment. (**c**) An illustration of spurious vector judgment.

*3.4. Evaluation Method*

To quantitatively evaluate the difference in magnitude and that in direction between the GOCI inversion SSC and measured SSC, we use the average angular error (AAE) and average relative magnitude error (AME) by Chen [40] to verify the inversion results. Herein, the direction and velocity of the SSC default to two-dimensional $s = (u, v)$, without considering the current in the vertical direction. The angle error and relative amplitude error between the measured velocity $v_{buoy}$ and satellite inverted velocity $v_{inv}$ can be written as follows:

$$\left\{ \overline{\Delta\theta}, \overline{\Delta V / V} \right\} = \frac{1}{N} \sum\nolimits_{i,j} \left\{ \arccos\left\langle \frac{v_{inv}.v_{buoy}}{|v_{inv}||v_{buoy}|} \right\rangle, \frac{|v_{inv} - v_{buoy}|}{|v_{buoy}|} \right\} \quad (6)$$

The relative magnitude error is a dimensionless quantity. If $v_{buoy} = 0$, then $\overline{\Delta V / V} = 1$; and if $v_{inv} = v_{buoy} = 0$, then $\overline{\Delta V / V} = 0$. The AAE (AAE $= \overline{\Delta\theta}$) and AME (AME $= \overline{\Delta V / V}$) are used to quantitatively evaluate the results of satellite inversion and model calculation.

**4. Results**

*4.1. Vector Processing Results Based on the Multi-Correlation Coefficient Algorithm*

Figure 7 shows a group of daily surface vectors obtained by the multi-correlation coefficient algorithm with TSM as the background on 13 August 2013; the rotation direction

is clockwise. In the period of 11:30–12:30 local time, the vector under the first correlation coefficient had inversion error, and there was no significant difference in the three vectors obtained in the remaining period, which confirms the existence of the mismatch.

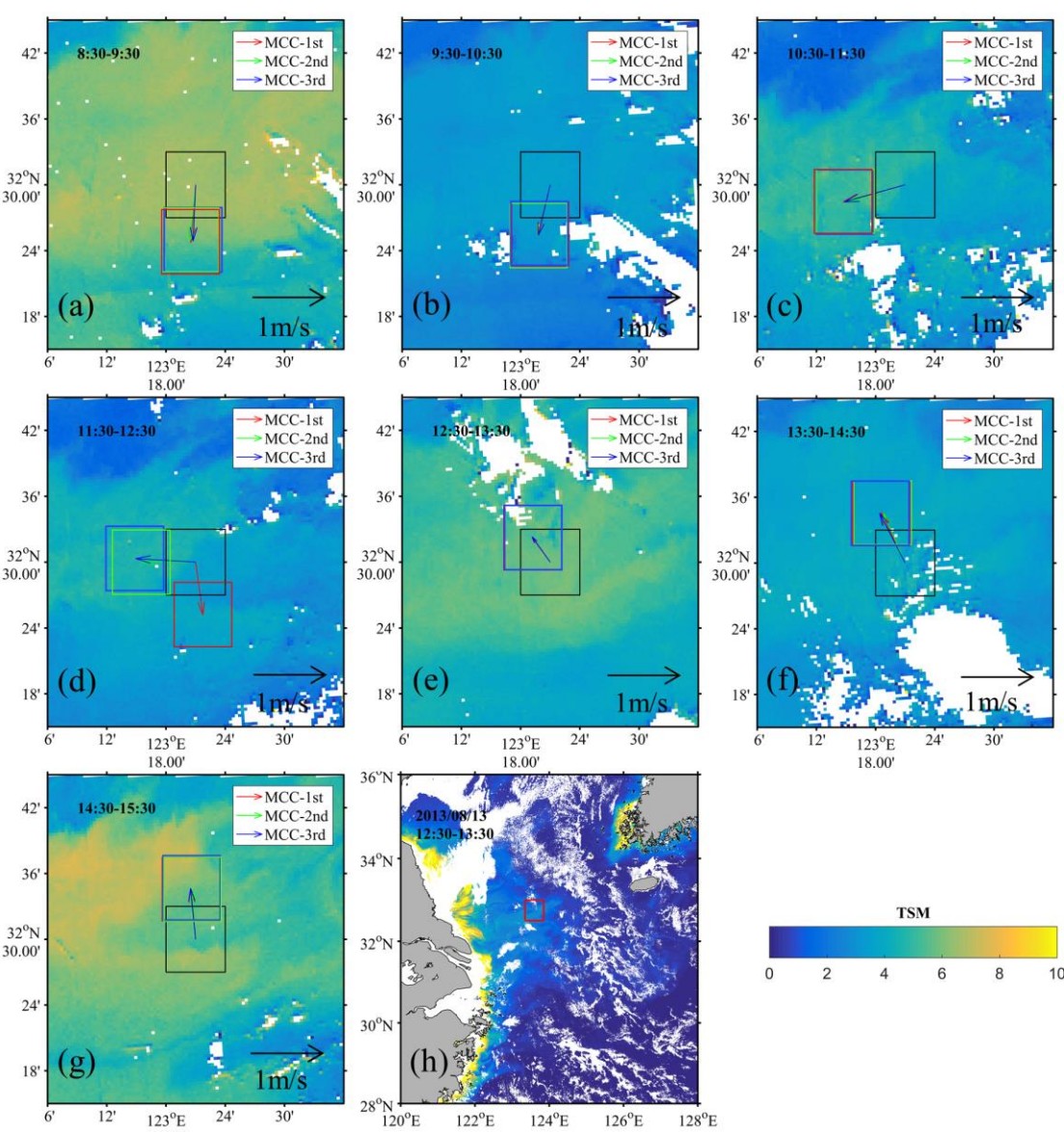

**Figure 7.** A group of daily surface vectors obtained by the multi-correlation coefficient algorithm with TSM as the background on 13 August 2013. The three colored vector arrows and boxes in (**a**–**g**) correspond to the SSC vectors and matching windows obtained under the first three correlation coefficients, respectively. The red box in (**h**) is where subplots (**a**–**g**) are located.

We selected 35 sets of case data, as shown in Figure 8b. Our method eliminates the spurious vectors in seven daily vectors (black dashed arrows in Figure 8b). As illustrated in Figure 8b, the GOCI vectors under the first three correlation coefficients obtained by the MCC algorithm are different in some time periods, and the spurious vectors under the 1st correlation coefficient can be identified by the established vector rotation direction in the OSU model. However, the case data used for evaluation should meet the following requirements: the selection of measured data should meet the corresponding GOCI data to obtain good satellite images within eight hours of each day. During the life cycle of the drifting buoys, we found that GOCI images on 10–11,13 August 2013, in the East China Sea and those on 27 June and 11,16 July 2012, and on 4–6 August 2012, in the Yellow Sea

were of high quality due to the relatively small cloud cover extent (Figure 1). Due to the complexity of the current system in the East China Sea, the current speed and direction will be quite different at different locations at the same time. Therefore, the selection of case data should meet the selected case consistent with the measured data, and should be as close as possible to the measured point position. In summary, we selected nine sets of case data in different sea areas to evaluate the new method. Figure 8a shows the distribution of measured data points and case data points. The case data that can be used for evaluation are close to the corresponding buoy points. The processing results of vectors are shown in Figure 9a–c), corresponding to the cases of the East China Sea, southern Yellow Sea, and northern Yellow Sea, respectively; there are differences among the results of each case. For example, in the East China Sea (Figure 9a) and the southern Yellow Sea (Figure 9b), the rotational direction of the fitted tidal ellipse was clockwise in the GOCI matching day and counterclockwise in the northern Yellow Sea (Figure 9c); the angle deviation between the buoy vector and GOCI vector in the northern Yellow Sea area was large, and the current direction shows reversing current characteristics. Furthermore, there are also differences in the number of error vectors identified. The number of error vectors identified on 16 July and 6 August was the 5th, 7th vector and the 1st, 6th vector, respectively; one vector was identified on the other days. Reflected from the statistics of average speed, the average speed of the SSC derived from the GOCI is 0.60 m/s, and the measured average speed of the drifting buoy is 0.58 m/s.

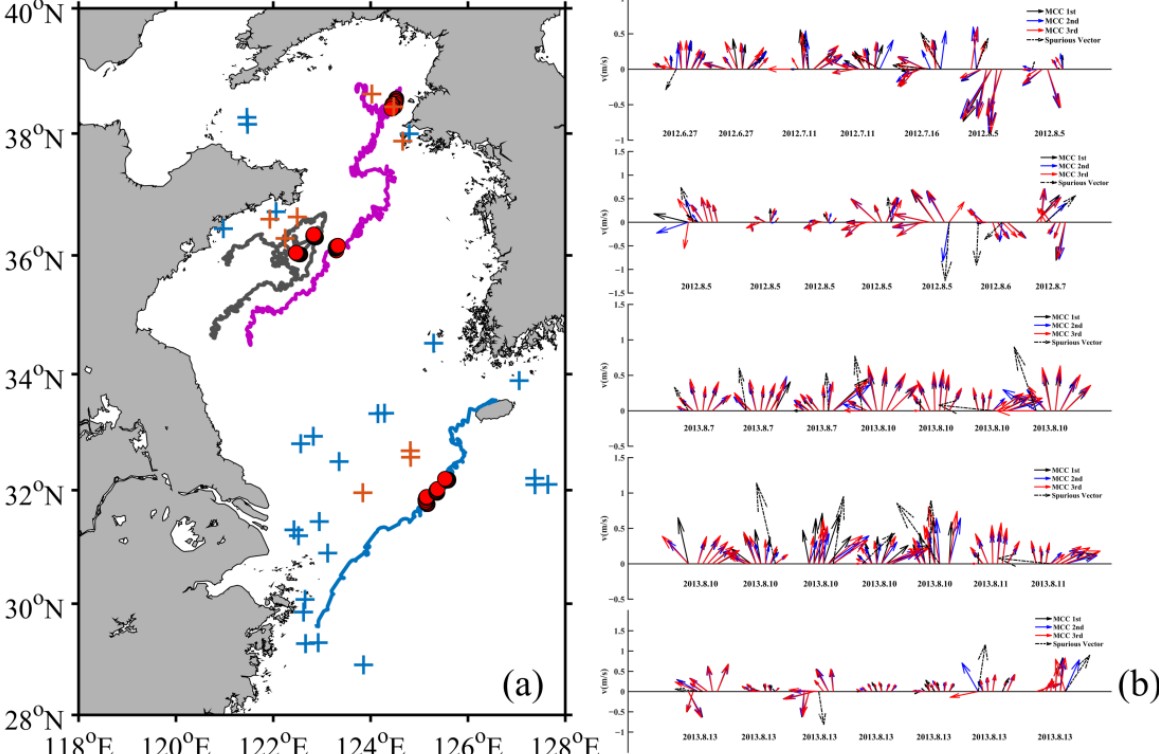

**Figure 8.** (**a**) Distribution of measured data and case data, where the red dots are the selected measured data points, the red crosses are the selected data case points, and the blue crosses are the remaining case points. (**b**) Comparison results of GOCI vectors under different correlation coefficients. Black represents the first vector, blue represents the second vector, red represents the third vector, and the dotted arrow represents the spurious vector.

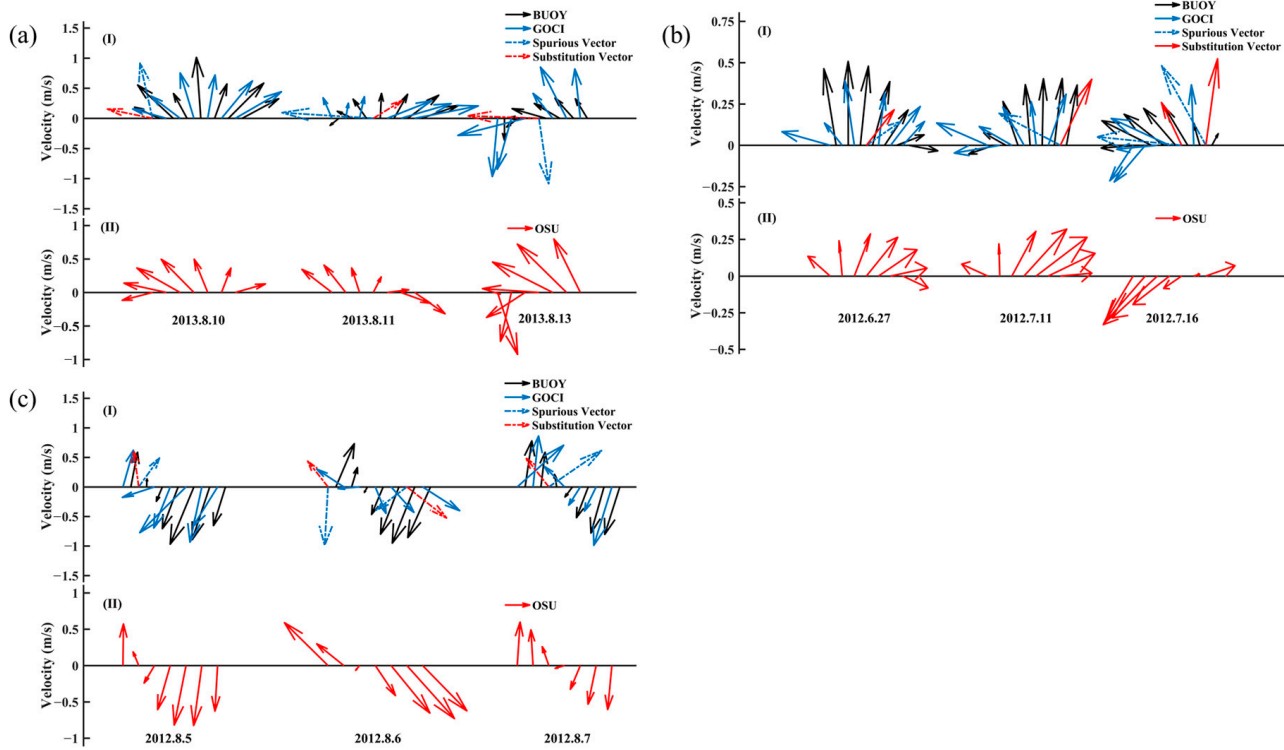

**Figure 9.** (**a**)-I showing the observed SSC vectors from the drifting buoy (BUOY; black arrows) and estimated from the satellite (GOCI; blue arrows) in the East China Sea. The blue dashed arrows represent the spurious vectors, and the red dashed arrows represent the substitution vectors. (**a**)-II showing the corresponding OSU tidal current vectors. The rotation direction can be determined. (**b**) Observed SSC vectors from drifting buoy and estimated from GOCI in the southern Yellow Sea. (**c**) Observed SSC vectors from drifting buoy and estimated from GOCI in the northern Yellow Sea.

### 4.2. Average Magnitude and Angular Error

To evaluate the results of the new method, we compared the method to the statistical data processing "angular limitation filter." Based on the rotation law of the tidal current ellipse, the angular limitation method considers the rotation direction and appropriate angle constraint, and then identifies and eliminates spurious vectors. Table 1 shows the AAE and AME values of the original 9-day data and the method after processing values during drifting buoy observations. The 9-day-average AAE value is 37.82°, and the AME value is 0.57 in the original data. In Chen [26] et al., the AAE and AME values are 51.76° and 0.50, respectively. Both methods have improved the results. The improvement of "multi-correlation coefficient optimization" for the AME value is better than that of the "angle limitation filter," and the averaged AAE value decreased to 30.28°, which means that the improvement is approximately 20%. The average AAE value obtained by the angular limitation filter has decreased to 34.56°, which means the improvement is approximately 9%. Comparison of the two filtering methods shows that the multi-correlation coefficient optimization method is better than the angular limitation filter method. Using the T-test method, we take the AME value and AAE values in Table 1 as samples to test whether the results of the two methods are statistically different from the original data. It is found that there is no significant difference between the data processed by the two methods and the original data, which may be due to the small proportion of spurious vectors. However, compared with the "angular limitation filter", the difference between the data processed by the new method and the original data is slightly larger, indicating that the new method contributes a further improvement. The AAE and AME results of the multi-correlation coefficient optimization method show that the direction and speed of the GOCI data are

similar to those of the measured buoy trajectory. Therefore, the MCC method is reliable for inverting tracer concentration to obtain the SSCs offshore of China.

**Table 1.** Results of different methods and original data for AAE and AME at different times.

| Area | Time | Original Data | | Angular Limitation Filter | | Multi-Correlation Coefficient Optimization | |
|---|---|---|---|---|---|---|---|
| | | AME | AAE (°) | AME | AAE (°) | AME | AAE (°) |
| ECS | 10 August | 0.27 | 18.83 | 0.26 | 13.35 | 0.23 | 13.43 |
| | 11 August | 0.34 | 55.01 | 0.25 | 49.42 | 0.26 | 49.84 |
| | 13 August | 0.78 | 33.16 | 0.86 | 19.42 | 0.75 | 18.92 |
| SYS | 27 June | 0.34 | 40.46 | 0.37 | 40.08 | 0.33 | 38.14 |
| | 11 July | 0.61 | 34.89 | 0.62 | 26.93 | 0.61 | 25.84 |
| | 16 July | 1.61 | 42.87 | 1.50 | 46.99 | 1.45 | 30.39 |
| NYS | 5 August | 0.40 | 16.66 | 0.40 | 11.26 | 0.41 | 14.08 |
| | 6 August | 0.47 | 66.88 | 0.47 | 72.31 | 0.52 | 54.83 |
| | 7 August | 0.35 | 31.59 | 0.40 | 31.25 | 0.38 | 27.06 |
| Average | | 0.57 | 37.82 | 0.57 | 34.56 | 0.55 | 30.28 |

Note: AAE: average angular error; AME: average relative magnitude error; ECS: East China Sea; SYS: southern Yellow Sea; NYS: northern Yellow Sea.

*4.3. OSU Tidal Model Data Evaluation*

Both the SSC obtained by GOCI inversion and that obtained from field measurements include periodic tidal currents and mean currents (residual currents), but the OSU tidal model can only give tidal currents. Note that tidal currents are an important part of dynamic processes in the East China Sea. Some previous studies have also shown that wind-driven velocities play a very limited role in the total SSCs in the Chinese seas, where tidal currents significantly dominate [32,41]. Therefore, we conjecture that tidal currents are the main factor driving changes in SSC during consecutive hours. Figure 10a compares continuously measured data and OSU mode data; the two have good consistency in the rotation direction. Figure 10b compares the measured current field and its corresponding OSU model current field during the active GOCI period; these are distributed at four different locations in the study area. In a few consecutive hours, the direction of each time period and the rotation direction of the OSU current vector were basically consistent with the measured current vector. In order to further verify and analyze the calculation results of OSU tidal model, the measured buoy data are selected to compare the calculated tidal currents value with the measured data. Table 2 shows the comparison between the measured results and the OSU model results. The measured average current speed is 0.39 m/s, which is close to the average current speed obtained by the OSU model of 0.38 m/s; and the average angle deviation is 43.96°. The current speeds of the two data samples were statistically tested, and the $p$ value was less than 0.05, indicating that the two groups of data come from different distributions, and there are significant differences between them, which is consistent with the fact that they belong to the measured data and the OSU model data, respectively. In summary, considering the measurement error, we can confirm our conjecture that in a few consecutive hours, the tidal current is the main factor driving the change in SSC. The tidal current results derived from the OSU model can be used to determine the rotation direction of the sea surface current and the direction of current in each period for several consecutive hours in the GOCI period.

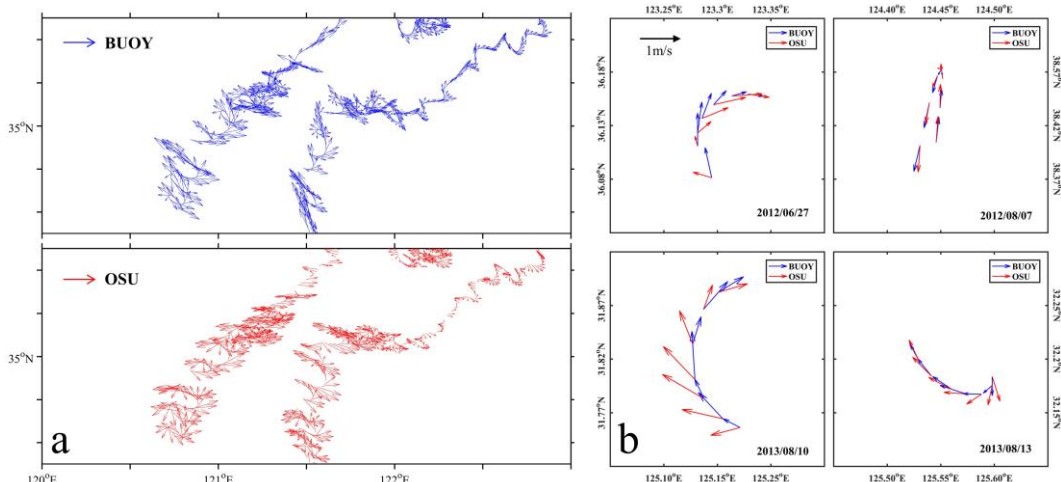

**Figure 10.** (**a**) Comparison results of partial continuous measured data and OSU model data in this study. (**b**) Comparison of buoy vectors and OSU vectors for some case data during the GOCI active period, which is 8:30–15:30 local time.

**Table 2.** Comparison of OSU mode calculation results and actual measured values.

| Buoy Number | Number of Sites | BUOY-ACS (m/s) | OSU-ACS (m/s) | AAE (°) |
|:---:|:---:|:---:|:---:|:---:|
| 1132711 | 1759 | 0.43 | 0.41 | 44.16 |
| 1131901 | 1787 | 0.28 | 0.34 | 49.90 |
| 1227890 | 320 | 0.45 | 0.38 | 37.82 |
| Average | 1289 | 0.39 | 0.38 | 43.96 |

Note: AAE: average angular error; ACV: average current speed.

### 4.4. SSC Mapping from GOCI and OSU

By using the MCC method and OSU tidal current model, we can obtain seven SSC maps each day. Figures 11 and 12 show the SSCs in the Yellow Sea obtained by the GOCI inversion (blue arrows) and the OSU model (red arrows) on 5 August 2012. The current field vectors obtained by the OSU model are relatively regular. There are spurious vectors in the current field retrieved by the GOCI inversion. The blank areas in the current field are due to the change in cloud amount or tracer concentration. The average speed obtained by the GOCI inversion is 0.63 m/s, and that obtained by the OSU model is 0.65 m/s. Comparing the current field maps obtained by the two methods, we find that the vector velocity of the current field decreases in central the Yellow Sea, presenting the characteristics of a weak current region; the current direction presents the characteristics of a north–south reversing current [42]. The currents derived by the MCC algorithm are compared with those in Ma et al. [43] and Hu et al. [5]; the currents calculated by the OSU tidal model are compared with those in Hu et al. [32] and Zhu et al. [44] The current field information is similar, and the current field is greatly affected by the counterclockwise tidal wave system in the Yellow Sea [45,46].

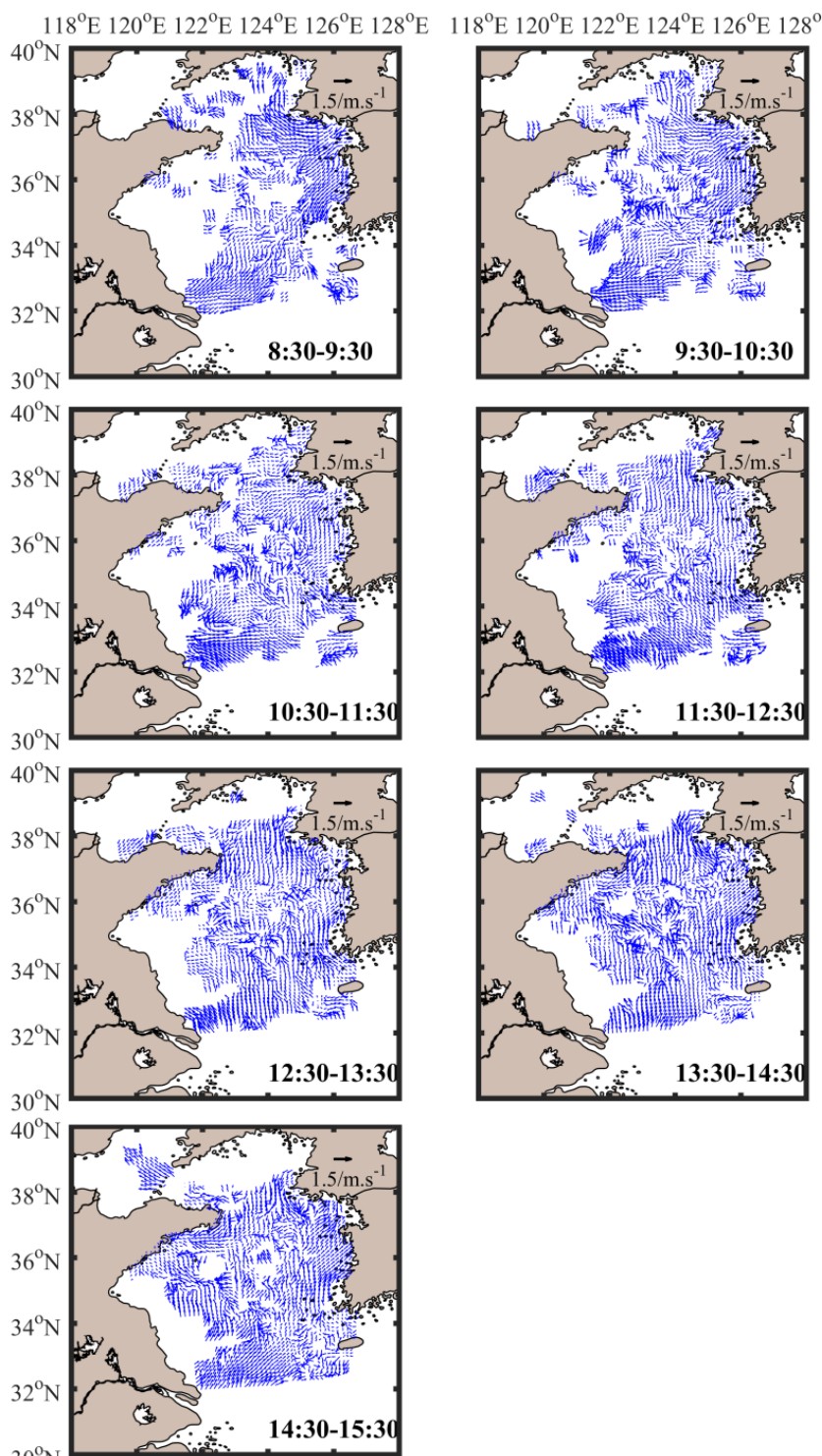

**Figure 11.** GOCI-derived SSC field in the Yellow Sea at seven intervals on 5 August 2012.

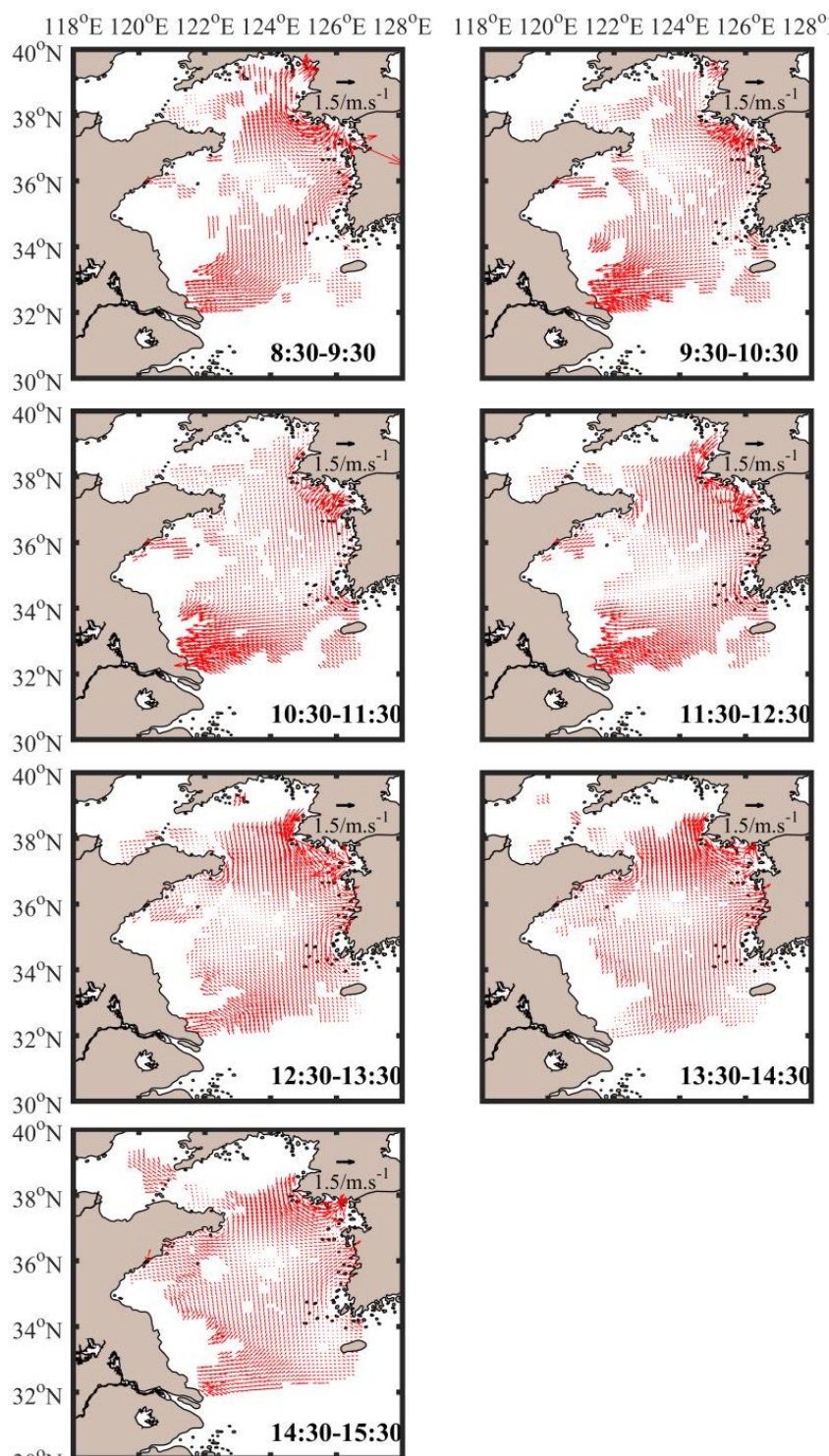

**Figure 12.** OSU-derived SSC field in the Yellow Sea at seven intervals on 5 August 2012.

## 5. Discussion

### 5.1. The Proportion of Accurate Vectors

Aiming at the mismatch phenomenon of the MCC algorithm in the SSC inversion in high turbidity areas, we propose to use the greatest three candidate vectors obtained by the multi-correlation coefficient algorithm as potential vectors. The rotation direction of the vectors within the tidal oscillation is used to identify and substitute the spurious vector, to improve the accuracy of SSC inversion. The method was verified using drifting buoy data.

Figure 13a shows the vector proportion obtained under each correlation coefficient after the algorithm processing. We can see that the vector proportion under the 1st correlation coefficient is the largest, reaching 82.54%, indicating that the MCC algorithm has strong applicability for SSC field inversion in the research area. The proportions of vectors under the 2nd and 3rd correlation coefficients are 11.11% and 6.35%, respectively, which are smaller and show a decreasing trend compared with the 1st correlation coefficient. On the one hand, the multi-correlation coefficient algorithm can identify and substitute the spurious vector of SSC inversion; at the same time, it also shows that when using this algorithm, considering the effect of computational efficiency and accuracy improvement, it is better to obtain the greatest three candidate vectors as a potential vector. The vectors under the 2nd and 3rd correlation coefficients are smaller than the replacement vectors. The distributions of the two in each period are shown in Figure 13b. It can be seen that the first and last period vectors account for a large proportion, indicating that the GOCI inversion had more error vectors in these two periods. The reason may be that the solar zenith angle during dawn and dusk is larger and less water color information is obtained by satellite sensors, resulting in poor inversion results [47]. In this study, since the spurious vectors are distributed in the study area with the case data (red and blue crosses in Figure 8a), the spatial distribution of the corresponding replacement vector is the same as the case data. It can be seen that there are more case data distributed in offshore areas, indicating that there are more wrong inversion results in the offshore areas.

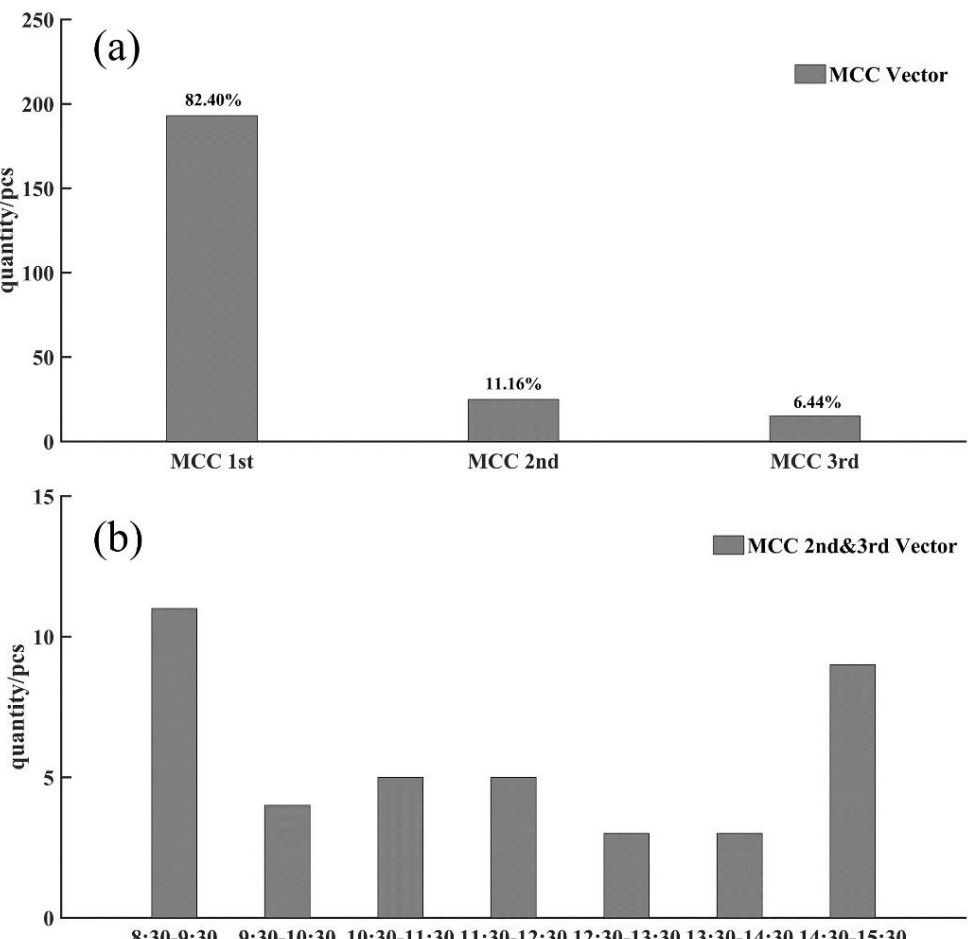

**Figure 13.** (**a**) The vector proportion obtained under each correlation coefficient after algorithm processing. MCC 1st, MCC 2cd, and MCC 3rd represent the vectors obtained under the first, second and third correlation coefficients, respectively. (**b**) The proportion of replacement vectors in each period after multi-correlation coefficient algorithm processing.

*5.2. Window Size Selection*

When using the MCC method to invert the SSC field, the selection of window size will affect the SSC inversion. The selection of the window should not only satisfy the requirement that "template window" should include enough spatial structure information of feature quantity (TSM), and that "search window" must cover the maximum moving distance of feature quantity, but also ensure that the obtained SSC field possesses better spatial consistency [20,48]. In this study, we used the area of 122.5°–125.5°E, 34.5°–37.5°N at 10:30–11:30 am on 5 August 2012 (less cloud cover, better inversion effect). For the case study, different window sizes were selected to compare and analyze the inversion results of the respective current fields. As shown in Table 3, W1-W7 are the different parameter settings of the seven window sizes used in the MCC method. For example, W1 represents the template window size of 10 × 10 pixels and the search window size of 24 × 24 pixels. Figure 14 shows the SSC vector diagram obtained under each parameter. Parameters W1, W2 and W4 have better inversion effects, but the current speed obtained by W2 is too large, and the current field obtained by W3 has obvious inversion error. Table 4 shows the SSC characteristics and vector coverage of the current field obtained under each parameter, where the vector coverage is the ratio of the actually obtained vector number to the theoretically obtained vector number. We can see that the average speeds obtained by W1, W4 and W7 are close to the measured average speed of 0.45 m/s, but the coverage rate of W7 is lower. Although the results obtained by W2 have a large coverage rate, the average current speed is relatively large, the maximum current speed reaches 2.95 m/s, and the error is relatively large. For the spatial consistency test of the obtained vectors, we use the method of neighborhood requirements [27]; that is, to compare the current speed difference and angle deviation between the target vector and its adjacent vector (up to eight, each direction includes an adjacent vector of the diagonal), and use the method mentioned in Section 3.4 for evaluation. Here, we stipulate that the number of adjacent vectors around the target vector should not be smaller than 5. As shown in Figure 14h), I-III are the positions of the three target vectors selected from south to north in order. Table 5 shows the AME and AAE values of the three target vectors and their adjacent vectors under each parameter. The AME and AAE values of the target vectors obtained by W1, W2, W4 and W5 are smaller than those of the adjacent vectors and have better spatial consistency. In summary, the current field obtained by selecting the window with the parameter size of W1 or W4 is better. Taking all the factors into account, we decided that W4 should be used as the standard setup for application of the MCC on GOCI-derived image pairs with T = 1 h.

**Table 3.** Different parameter settings of seven window sizes.

| MCC | W (=H) | W1 | W2 | W3 | W4 | W5 | W6 | W7 |
|-----|--------|----|----|----|----|----|----|----|
| $T_{sub}$ | pixels | 10 | 10 | 20 | 20 | 28 | 20 | 28 |
| $S_{sub}$ | pixels | 24 | 36 | 24 | 36 | 36 | 48 | 48 |

Note: W(=H): width(=height); $T_{sub}$: template window; $S_{sub}$: search window.

**Table 4.** SSC characteristics and vector coverage of SSC field under different window sizes.

|  | W1 | W2 | W3 | W4 | W5 | W6 | W7 |
|--|----|----|----|----|----|----|----|
| Max-speed (m/s) | 1.11 | 2.95 | 1.57 | 1.39 | 1.18 | 2.95 | 1.46 |
| Min-speed (m/s) | 0.23 | 0.36 | 0.79 | 0.22 | 0.01 | 0.15 | 0.05 |
| Ave-speed (m/s) | 0.60 | 1.09 | 1.00 | 0.68 | 0.28 | 0.74 | 0.51 |
| PCV (%) | 86.54 | 97.65 | 56.20 | 96.58 | 70.09 | 87.61 | 77.56 |

Note: Ave-speed: average speed; PCV: percentage coverage values.

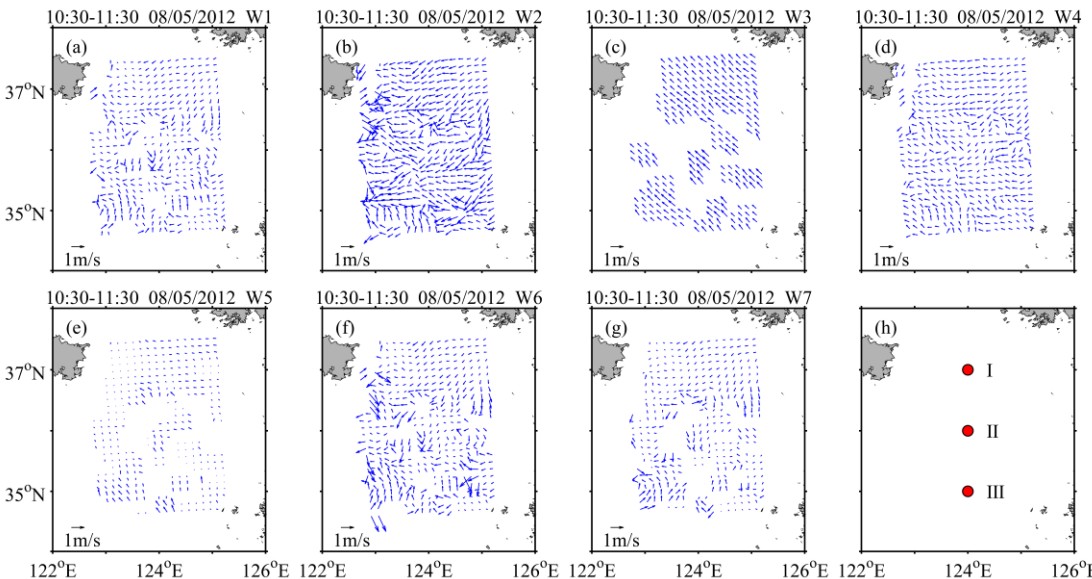

**Figure 14.** (**a**–**g**): SSC vectors obtained under each window size parameter. Red dots in (**h**) are the position of the target vectors selected.

**Table 5.** AME and AAE values of the three target vectors and their adjacent vectors under each window size.

| Target Vector | W1 | | W2 | | W3 | | W4 | |
|---|---|---|---|---|---|---|---|---|
| | AME | AAE(°) | AME | AAE(°) | AME | AAE(°) | AME | AAE(°) |
| I | 0.07 | 10.43 | 0.10 | 9.70 | 0.09 | 2.08 | 0.09 | 13.63 |
| II | 0.24 | 54.72 | 0.36 | 42.48 | —— | —— | 0.76 | 39.24 |
| III | 0.07 | 16.70 | 0.26 | 29.73 | —— | —— | 0.45 | 29.94 |
| Average | 0.13 | 27.28 | 0.24 | 27.31 | —— | —— | 0.43 | 27.60 |

| Target Vector | W5 | | W6 | | W7 | |
|---|---|---|---|---|---|---|
| | AME | AAE(°) | AME | AAE(°) | AME | AAE(°) |
| I | 0.20 | 5.21 | 0.10 | 13.18 | 0.13 | 11.76 |
| II | 6.61 | 59.76 | 0.69 | 56.20 | 0.38 | 20.41 |
| III | 0.46 | 23.16 | 0.49 | 32.10 | —— | —— |
| Average | 2.42 | 29.37 | 0.43 | 33.83 | —— | —— |

### 5.3. Condition Analysis of Current Detection

The GOCI-derived current field is affected by conditions such as cloudiness and tracers. For the analysis of the conditions for effective detection of SSC, we attempted to change the types of tracers and compare the differences in the GOCI current field derived from different tracers. Table 6 and Figure 15 show the statistical and comparative results of the SSC field derived by the GOCI inversion using TSM, Chl-a, and remote sensing reflectance (Rrs) at 555 nm as tracers at noon in the central Yellow Sea with a better inversion effect. The currents derived from different tracers are different from each other. The table shows that TSM gives the best effect in selected cases and obtains the largest number of current field vectors.

**Table 6.** Statistics of the GOCI current field inversion results of three different tracers.

| Date | Time | Chl-a | | | Rrs | | | TSM | | |
|---|---|---|---|---|---|---|---|---|---|---|
| | | Number of Vectors | AME | AAE (°) | Number of Vectors | AME | AAE (°) | Number of Vectors | AME | AAE (°) |
| 27 June | 11:30–12:30 | 1010 | 1.13 | 21.83 | 955 | 1.22 | 27.04 | 1005 | 0.67 | 13.62 |
| | 12:30–13:30 | 976 | 1.90 | 27.15 | 952 | 0.54 | 16.17 | 981 | 1.59 | 18.31 |
| 11 July | 11:30–12:30 | 472 | 0.30 | 13.99 | 448 | 0.62 | 13.51 | 476 | 0.53 | 15.34 |
| | 12:30–13:30 | 580 | 0.25 | 6.44 | 487 | 0.50 | 16.10 | 553 | 0.52 | 12.54 |
| 16 July | 11:30–12:30 | 467 | 0.32 | 24.74 | 464 | 0.76 | 39.16 | 484 | 0.32 | 29.59 |
| | 12:30–13:30 | 534 | 0.73 | 15.95 | 503 | 1.42 | 24.14 | 541 | 0.99 | 12.34 |
| Average | | 673 | 0.77 | 18.35 | 634 | 0.84 | 22.69 | 673 | 0.77 | 16.96 |

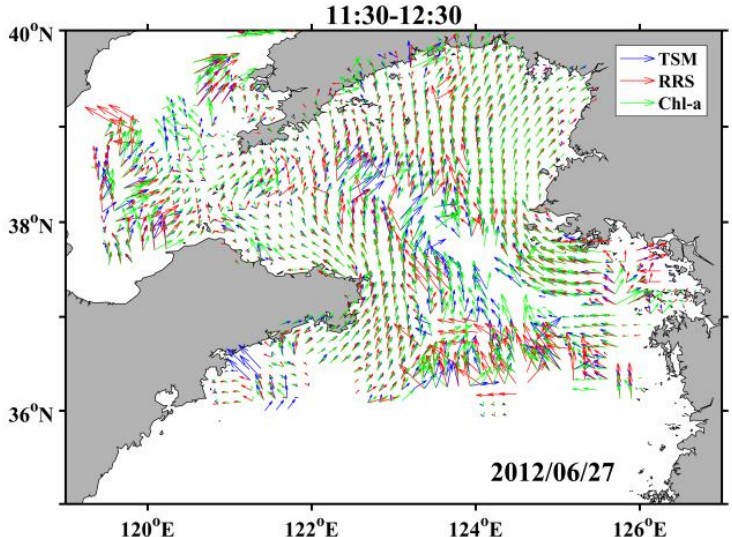

**Figure 15.** Comparison of GOCI-derived current field results of three different tracers.

## 6. Conclusions

In this study, we used the MCC algorithm to obtain the SSC field in the East China Sea and the Yellow Sea, with the TSM concentration data retrieved by GOCI remote sensing as the tracer. Comparison with the drifting buoy data shows that the method is suitable for SSC inversion offshore of China. The OSU model current field was consistent with the in situ measurement data current field in terms of the current direction and considered the image mismatch in the MCC method in the high turbidity water. We developed a new method, named the "multi-correlation coefficient optimization algorithm based on the OSU tidal current model" to improve the accuracy of the GOCI-derived time series data. This method considers the three greatest candidate acquisitions from multi-correlation coefficients as potential vectors. The rotation direction of the vectors within the tidal oscillation is used to identify and substitute for the spurious vector, which improves the inversion accuracy.

Compared with the existing spurious vector elimination methods, the AAE of the new method is reduced from 37.82° to 30.28°, with an improvement of approximately 20% and the AME, approximately 4%. The angle limitation method AAE decreased from 37.82° to 34.56°, with an improvement of approximately 9%, with no significant change in AME. Based on visual and quantitative results, our new method improves the accuracy and ensures the integrity of the data compared with other state-of-the-art vector filtering methods.

In addition, we also count the proportion of accurate vectors after the algorithm processing and conclude that it is better to obtain the three maximum three candidate vectors as potential vectors. For the selection of window size in the MCC method, we

evaluated the current characteristics derived under various parameters, and selected the optimal window size. For the selection of tracers, the TSM-derived current field gave higher accuracy and more vectors than the other two products.

**Author Contributions:** Conceptualization, J.C. and H.C.; methodology, J.C. and H.C.; software, H.C.; data analysis, J.C. and H.C.; investigation, J.C. and H.C.; resources, Z.C., H.H. and F.G.; writing H.C. and J.C.; supervision, J.C.; project administration, J.C.; funding acquisition, J.C. All authors have read and agreed to the published version of the manuscript.

**Funding:** This research was funded by the Project of State Key Laboratory of Satellite Ocean Environment Dynamics, Second Institute of Oceanography (No. SOEDZZ2203), NSFC Zhejiang Joint Fund for the Integration of Industrialization and Informatization (Grant U1609202), the National Key Research and Development Program of China (Grant 2016YFC1400903), and National Natural Science Foundation of China (Grants 42076216, 41376184 and 40976109).

**Data Availability Statement:** The authors thank the Korea Institute of Ocean Science & Technology (KIOST) for providing the freely available GOCI data (http://kosc.kiost.ac.kr/, accessed on 13 July 2021) and Oregon State University for providing the tidal current predictions (http://www.topx.net, accessed on 13 July 2021). The drifting buoy data (SIO; August 2013, June–August 2012) for this study are available from Cao (http://www.dx.doi.org/10.11922/sciencedb.730, accessed on 11 December 2020).

**Conflicts of Interest:** The authors declare no conflict of interest.

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
