# Peer review of "A Novel Multi-Candidate Multi-Correlation Coefficient Algorithm for GOCI-Derived Sea-Surface Current Vector with OSU Tidal Model"

_remotesensing, doi:10.3390/rs14184625_

Round 1

Reviewer 1 Report

I made a search and found a recent article from 2020 that uses a similar methodology and a similar geostationary optical sensor, which has not been considered in the introduction and/or the discussion of results sections. Please check the article from

 Zhu, Z., Geng, X., Li, S. et al. Ocean surface current retrieval at Hangzhou Bay from Himawari-8 sequential satellite images. Sci. China Earth Sci. 63, 1026–1038 (2020). https://doi.org/10.1007/s11430-019-9557-7.

 Zhu and co-authors report satellite current estimations with a similar approach, the geostationary platform is different, but the sensor is similar AHI on board the Himawari-8, in fact there are comparisons between GOCI and AHI for different satellite geophysical products.

Articles included in the introduction section are from the 80s and 90s and you haven’t included this previously mentioned article. I am sure that a discussion on the results could make your paper more interesting considering the different platforms and sensors. 

On the other hand, if sea surface currents are being indirectly estimated through the movement of TSM spatial patterns, geometric accuracy should be also taken into account for the GOCI sensor, there is a recent article on this. At least, the issue should be mentioned and discussed. 

https://opg.optica.org/oe/fulltext.cfm?uri=oe-28-5-7634&id=427945

Reviewer 2 Report

Just a few notes:

- The background with all the previous studies of currents and the MCC is impressive!

- Figure 1 and Figure 2 showing the path of the buoy and the buoy itself are very helpful.

- on line 141, I appreciate the use of Beijing time, as it gives an idea of what the local daylight conditions are.  This is quite important for satellite data.

- on line 162 it says "preprocessed in terms of atmospheric correction and mask" --  I believe this should be "masks" or "masking" as the singular word mask does not quite feel right given that there are multiple masks applied to ocean color data.

- the explanation and presentation of the formulas along side Figure 3 is also quite appreciated and makes it more approachable to people without a background in current modeling.

Over all, I felt the presentation was well done and polished through-out.  While I am not as familiar with the field of current vectors as I would like, I was able to understand and learn from the experiments done in this paper.  Well done -- and having accurate current information in highly turbid water is indeed quite important.

Reviewer 3 Report

The authors introduce an improvement of the maximum cross coefficient, called "Multi-Candidate Multi-Correlation Coefficient Algorithm". The improvement is not an introduced property. The main problem lies in the insufficient verification data and weak text showing the method. Please respond to all my comments by adding or removing paragraphs in the manuscript. 

Main questions:

The model is verified using an extremely small number of data/days. The authors need to demonstrate that the selected data are representative and that the data period has sufficient variability to support the implementation of the "new" algorithm.

The manuscript is fuzzy. The authors need to rewrite the manuscript to focus on the main results what is introducing multi-correlation coefficient optimization algorithm. 

Are the values reported in Table 1 and Table 2 statistically significantly different from each other? This needs to be tested. 

The authors conclude that the proportion of the first vector is the most important. The authors do not introduce spatial/temporal characteristics of the vector proportion under the 2nd and 3rd correlation coefficients. This should be corrected. 

Verification of the OSU model with the in situ data is not important for the manuscript. The verification is not done because there is very little in-situ data. This should be corrected in the conclusion.

The title of the manuscript is not supported by the results as the method is not presented correctly. It is a case study analysis. 

Key Comments:

L153 Please check the link www.topx.net. It is a very wrong link!

Did the authors create the L2 TSM product? If not, please remove paragraph L161-164. Equation 1 and paragraphs L171-L171 are common knowledge, so remove them from the manuscript.

Please introduce the template matching technology (L168).

Equation 3. what is the range of arctanges? (-pi/2 - pi/2)? How do the authors arrive at degrees between 0 and 2pi?

The MCC method must be introduced to support "template window" and "search window".

Section 3.2 is poorly written. Please rewrite it and add more information. How is equation 4 related to the Pythagorean theorem? What is equation 5. 

Not introduced: " first three correlation coefficients of GOCI". Please explain.

Equation 6 is an equation from Chen [38] and should therefore be removed from the manuscript. 

L324 "Angular limitation filter" is introduced in the results chapter. Please add the paragraph in the method section.

L353 The authors hypothesise based on a figure showing 4 case studies ("It is evident from the figure that the direction of rotation of the OSU current vector and the measured current vector essentially coincide in each period within several consecutive hours"). This is not a correct scientific conclusion. Delete paragraph 4.3 or make a correct analysis. 

Are the differences between BOUY-ASV and OSU-ASV reported in Table 2 statistically significant? Verify statistical significance by performing a statistical test. 

Please write how circular vector analysis is applied. How is the circular correlation calculated?

L337 What does "relatively regular" mean?

L422 needs to be introduced in the data section. Authors need to clearly state what the spatial/temporal range of the data is, and the windows in the analysis.
